# Nanomechanical characterization of quantum interference in a topological insulator nanowire

Minjin Kim[1], Jihwan Kim[2], Yasen Hou[3], Dong Yu[3], Yong-Joo Doh [4], Bongsoo Kim[1], Kun Woo Kim[5]* & Junho Suh[2]*

Aharonov–Bohm conductance oscillations emerge as a result of gapless surface states in topological insulator nanowires. This quantum interference accompanies a change in the number of transverse one-dimensional modes in transport, and the density of states of such nanowires is also expected to show Aharonov–Bohm oscillations. Here, we demonstrate a novel characterization of topological phase in $Bi_2Se_3$ nanowire via nanomechanical resonance measurements. The nanowire is configured as an electromechanical resonator such that its mechanical vibration is associated with its quantum capacitance. In this way, the number of one-dimensional transverse modes is reflected in the resonant frequency, thereby revealing Aharonov–Bohm oscillations. Simultaneous measurements of DC conductance and mechanical resonant frequency shifts show the expected oscillations, and our model based on the gapless Dirac fermion with impurity scattering explains the observed quantum oscillations successfully. Our results suggest that the nanomechanical technique would be applicable to a variety of Dirac materials.

[1] Department of Chemistry, Korea Advanced Institute of Science and Technology, Daejeon, Korea. [2] Quantum Technology Institute, Korea Research Institute of Standards and Science, Daejeon, Korea. [3] Department of Physics, University of California at Davis, Davis, CA 95616, USA. [4] Department of Physics and Photon Science, Gwangju Institute of Science and Technology, Gwangju, Korea. [5] Center for Theoretical Physics of Complex Systems, Institute for Basic Science (IBS), Daejeon, Korea. *email: kkimx4@ibs.re.kr; junho.suh@kriss.re.kr

The discovery of two-dimensional (2D) gapless Dirac fermions in graphene[1] and topological insulators (TI)[2] has sparked extensive ongoing research toward applications of their unique electronic properties[3–6]. The gapless surface states in three-dimensional (3D) insulators indicate a distinct topological phase of matter characterized by a non-trivial $Z_2$ invariant[6,7] with time reversal symmetry that can be verified by angle-resolved photoemission spectroscopy[2,8,9] or magnetoresistance quantum oscillation[10,11]. In TI nanowires, the gapless surface states exhibit Aharonov–Bohm (AB) oscillations in conductance due to a change in the number of transverse one-dimensional (1D) modes in transport[12–15]. Thus, the density of states (DOS) of such nanowires is expected to show AB oscillation as well; however, magneto-oscillation of the DOS, which reflects the contribution of the topological surface states of TI nanowire, has yet to be observed.

Here, we adopt nanomechanical measurements[16–19] that reveal AB oscillations in the DOS of a topological insulator. The TI nanowire under study is an electromechanical resonator embedded in an electrical circuit, which couples mechanical resonance to the quantum capacitance originating from the finite DOS of the nanowire's surface states. The quantum capacitance effects from DOS oscillation modulate the circuit capacitance, thereby altering the spring constant to generate mechanical resonant frequency shifts. The resonant frequency of nanomechanical motion carries the signal that shows AB oscillations correlated to the conductance modulation, suggesting the application of our novel sensing scheme to studies of various topological materials with Dirac electronic structures. Detection of the quantum capacitance effects from surface-state DOS is facilitated by the small effective capacitances[20] and high quality factors[21,22] of nanomechanical resonators, and as such the present technique could be extended to study diverse quantum materials at nanoscale.

## Results

**Nanomechanical resonance and quantum capacitance**. We design our device to measure the fundamental mode frequency of flexural vibration in a suspended TI nanowire fabricated with single-crystalline $Bi_2Se_3$ (Fig. 1a). The mechanical motion of the nanowire changes the electrostatic energy stored in a circuit, from which shifts in resonant frequency can be computed. In the circuit in Fig. 1b, as DC gate voltage $V_g$ is introduced, charges are induced to compensate for the potential difference between the gate and nanowire. At the same time, nanowire chemical potential $\mu$ is changed by induced charge $Q$ as

$$V_g = \frac{Q}{C_G} + \frac{\mu - \mu_0}{e}, \tag{1}$$

where $C_G$ is the geometric capacitance between the nanowire and gate electrode, $\mu_0$ is the intrinsic chemical potential of a nanowire with $V_g = 0$, and $e$ is the elementary charge. In addition to $V_g$, a radio-frequency (RF) voltage $V_{RF}$ is applied to induce nanowire vibration, a motion that modifies $C_G$ followed by the modulation of $Q$ and $\mu$ according to Eq. (1). The amplitude of charge modulation measured as $V_{OUT}$ is maximized when $V_{RF}$ meets the mechanical resonant frequency[20]. In this way, we trace the mechanical resonant frequency by sweeping the gate voltage $V_g$ (Fig. 1c) or magnetic field $B$. Note that the mechanical vibration (~100 MHz) is much slower than the typical time scales of electron dynamics considered here (relaxation time ~1 ps, see Supplementary Note 1), and thus we assume the electrostatic equation to hold throughout.

If the density of states is infinite as in metals, chemical potential remains at $\mu_0$ regardless of $Q$. With the finite DOS of TI

nanowires, however, change in gate voltage is divided into the potential difference between gate and wire and the change in chemical potential, as

$$\delta V_g = \frac{\delta Q}{C_G} + \frac{\delta \mu}{e} = \delta Q \left( \frac{1}{C_G} + \frac{1}{e^2 L \nu} \right), \tag{2}$$

where $\nu = \delta\left(\frac{Q}{Le}\right)/\delta\mu$ is the number of electronic states per unit length per unit energy, and $L$ is wire length. From this relation, we introduce quantum capacitance to account for the potential difference made internally by the induced charge[16], $C_Q = e^2 L \nu$. In our experiments, we estimate that quantum capacitance dominates geometric capacitance ($C_Q/C_G \sim 10^3$, see Supplementary Note 2 and Note 3), and thus the first term in Eq. (2) accounts for the majority of gate voltage change. However, AB oscillation in the mechanical resonant frequency turns out to originate mostly from the second term related to the DOS in our experiment.

**The DOS and resonant frequency shift**. When a magnetic flux ($\Phi$) is applied along the nanowire axis, 1D sub-band energy is described by[12,15]

$$\varepsilon(n, k, \Phi) = \pm \hbar v_F \sqrt{k^2 + \frac{(n + 1/2 - \Phi/\Phi_0)^2}{R^2}}, \tag{3}$$

where $n$ is the sub-band index, $k$ is a 1D wave number along the nanowire direction, $h$ is Planck's constant ($\hbar = \frac{h}{2\pi}$), $v_F$ is the Fermi velocity[23] ($\approx 5 \times 10^5$ m/s for $Bi_2Se_3$), $R$ is nanowire radius, and $\Phi_0$ is the flux quantum ($= h/e$). In cylindrical geometry, a gapless Dirac fermion picks up the Berry phase of $\pi$ by encircling the perimeter, and a gapless 1D mode appears when an external magnetic field supplies an additional AB phase $\pi$. As an example, we draw the band dispersion of 1D modes for $\Phi = 0$ and $\Phi = \Phi_0/2$ in Fig. 1d; appearance of the 1D gapless mode is followed by an energy shift of neighboring transverse modes, thus also changing conductance $G$ at a finite energy (Fig. 1e).

The appearance of the gapless mode also affects the DOS (Fig. 1f), with related quantum capacitance effects resulting in mechanical resonant frequency shifts that can be obtained by taking the variation of electrostatic energy (Fig. 1g–h). Induced charge $Q$ is computed by integrating the DOS from the initial chemical potential $\mu_0$ to $\mu$, as $Q = Le \int_{\mu_0}^{\mu} \nu(E) dE$, where the sign of $Q$ is negative for $\mu < \mu_0$ (or $V_g < 0$). With mechanical vibration, change in electrostatic energy for a fixed gate voltage reads

$$\delta U_{ec} = \delta\left(\frac{Q^2}{2C_G}\right) - V_g \delta Q + \delta\left(Q\frac{\mu - \mu_0}{e}\right), \tag{4}$$

where the first, second, and third terms are from the energy stored in the geometric capacitance, work done to the $V_g$ source, and quantum capacitance charging energy, respectively. By taking the second derivative of $U_{ec}$ with respect to nanowire displacement $x$, we obtain the change in spring constant due to electrostatic energy by

$$k - k_0 \approx -\frac{1}{2}\left(\frac{\ddot{C}_G}{C_G^2}\right)Q^2 + \frac{e}{2}\left(\frac{\dot{C}_G}{C_G}\right)^2 \frac{\partial}{\partial \mu}\left(\frac{1}{C_Q^2}\right)Q^3 = k_I + k_{II}, \tag{5}$$

where the dots indicate derivatives with respect to $x$, $k_0$ is the bare effective spring constant of the mechanical resonator, and $C_G \ll C_Q$ is assumed (Supplementary Note 4). We are mainly interested in the modulation of resonant frequency with respect to magnetic field: $\Delta k_{I,II} = k_{I,II} - \langle k_{I,II} \rangle_B$ where $\langle \dots \rangle_B$ refers to the averaged value over the magnetic field at a given $V_g$.

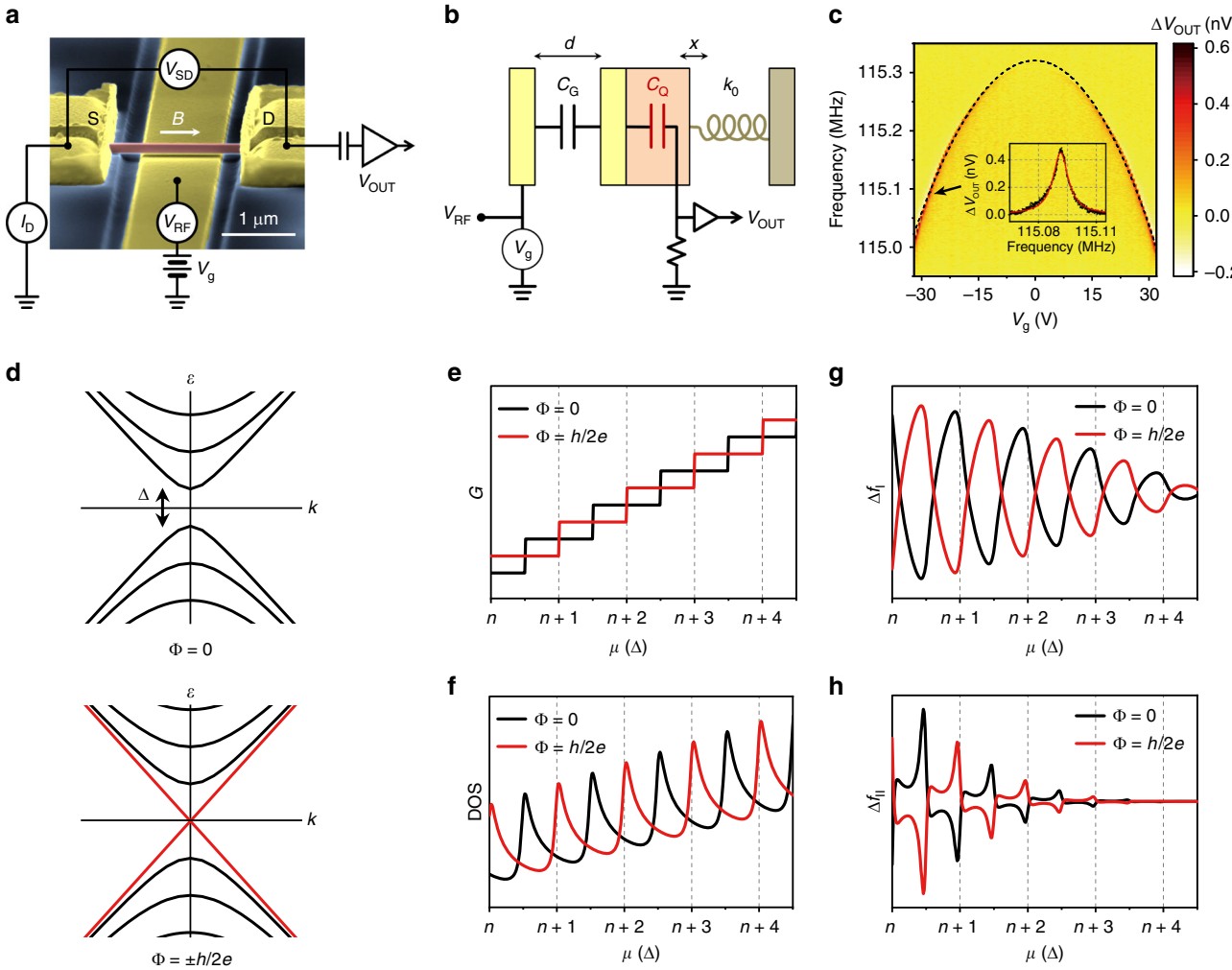

**Fig. 1** Device configuration and expected Aharonov–Bohm (AB) oscillations in conductance and mechanical resonant frequency. **a** Scanning electron microscope (SEM) image of the $Bi_2Se_3$ nanowire mechanical resonator. Electrical conductance is measured by comparing drain current ($I_D$) and source-drain voltage ($V_{SD}$). DC and radio-frequency (RF) voltages ($V_g$ and $V_{RF}$ respectively) are applied to the gate electrode, with RF voltage at the drain electrode amplified and recorded ($V_{OUT}$) to actuate and detect mechanical resonance. A magnetic field ($B$) is applied parallel to the nanowire axis to study AB oscillation due to TI surface states. **b** The equivalent circuit for RF frequency involves mechanically compliant geometric capacitance ($C_G$) and quantum capacitance ($C_Q$) of the nanowire. **c**, Colourmap of $V_{OUT}$ for a range of $V_g$. The parabolic shift in resonance is due to the effective spring constant change dominated by $C_G$. Inset: Example of mechanical resonance with a resonant frequency of 115.09 MHz and quality factor of $1.2 \times 10^4$ at $V_g = -28.0$ V. The solid red line is a Lorentzian fit. **d** Energy dispersion of surface states in a TI nanowire when magnetic flux $\Phi$ is zero or $\Phi_0/2$ where $\Phi_0$ is the flux quantum ($=h/e$). Red lines indicate the non-degenerate gapless mode. **e**, **f** Expected conductance ($G$) (**e**) and density of states (DOS) (**f**) of the surface states as a function of chemical potential $\mu$ in units of 1D sub-band gap $\Delta$, when the magnetic flux is zero (black) or $\Phi_0/2$ (red). **g**, **h** Mechanical resonant frequency shifts ($\Delta f_0 = \Delta f_I + \Delta f_{II}$) as a result of DOS oscillation via quantum capacitance effects. The resonant frequency shift is calculated from Eq. (5) considering DOS oscillation in (**f**), and its two terms $\Delta f_I$ (**g**) and $\Delta f_{II}$ (**h**) are proportional to $Q^2$ and $Q^3$, respectively

Mechanical resonant frequency shifts corresponding to $\Delta k_{I,II}$ are given by $\Delta f_{I,II} = \left(\frac{\Delta k_{I,II}}{2\langle k \rangle_B}\right) \cdot \langle f \rangle_B$. In Eq. (5), $k_I$ is easily understood as the gate capacitance effect[24] with a correction for induced charge due to the magnetic flux and chemical potential. Otherwise, the second term $k_{II}$ indicates a novel effect particularly important to our 1D TI system, as it contains a derivative of the DOS (or quantum capacitance) with respect to chemical potential, as well as the third-order power of induced charge. Considering that $Q$ is negative (i.e., $V_g < 0$, where our extensive measurements are conducted), the DOS derivative in $\Delta k_{II}$ makes an important contribution (Fig. 1h): as the chemical potential approaches the Dirac point where the DOS profile gets sharper with less scattering, $\Delta f_{II}$ grows significantly faster than $\Delta f_I$. Accordingly, at higher energies with larger DOS, $\Delta k_I$ dominates.

**AB oscillations in the conductance**. We first characterize the AB conductance oscillation[12–15] in our TI nanowire by measuring conductance at various magnetic fluxes and gate voltages (Fig. 2a). As indicated in the fast Fourier transformation (FFT, Fig. 2a, inset), the magneto-conductance $\Delta G$ oscillates with a period of $\Delta B = 0.4$ T, which is consistent with the expected period, $\frac{\Phi_0}{S} = 0.37$ T with nanowire cross-section area $S$ (Supplementary Note 5). There is only a small fraction of Altshuler–Aronov–Spivak oscillation with $\frac{\Phi_0}{2}$ period, indicating that the nature of transport is close to ballistic, rather than diffusive; the fact that AB oscillation amplitude is smaller than the conductance unit could originate from the presence of incoherent scatterings[15,25]. Note that the pronounced peak at $B = 0$ is the signature of weak anti-localization[13,26]. Conductance modulation occurs with gate voltages as well, and correlation between two

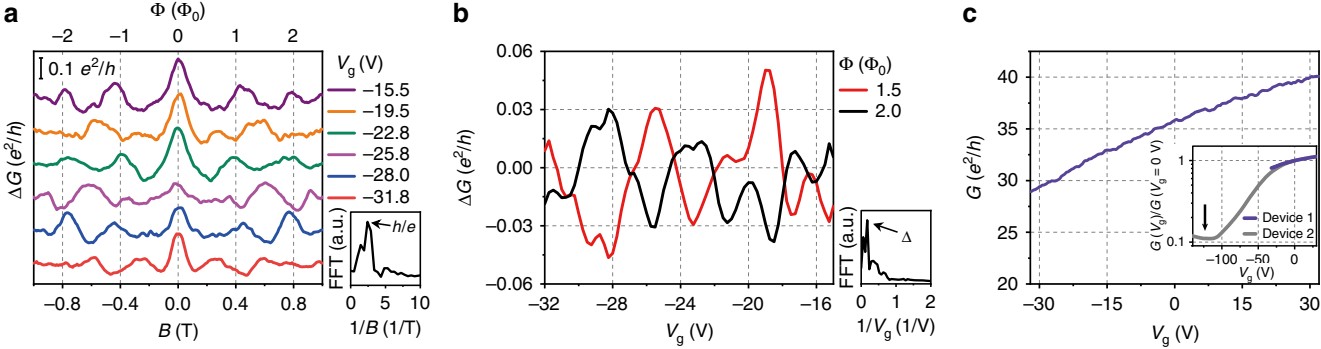

**Fig. 2** Conductance oscillation as a function of magnetic field and gate voltage. **a** Conductance oscillations as a function of magnetic field at different gate voltages after subtracting background conductance. The $\Delta G$ oscillations have a period of $\Delta B = 0.4$ T as identified from the first peak in the fast Fourier transformation (FFT) spectrum (inset). $\Delta B$ corresponds to one flux quantum (= $h/e$) across the nanowire cross-section, confirming AB oscillation. The magneto-conductance oscillations show periodic alternations with respect to gate voltages, as expected from the Berry phase in the surface-state Hamiltonian (Eq. (2)). Inset: FFT spectrum of $\Delta G$ vs. $B$ at $V_g = -22.8$ V. Arrows indicate the peak position at period $\Delta B = 0.4$ T. **b** $\Delta G$ as a function of gate voltage $V_g$ at two representative magnetic fluxes. The red line is the half-integer flux quantum ($B = 0.6$ T) and the black line is the integer flux quantum ($B = 0.8$ T). The period of the $\Delta G$ oscillations with respect to gate voltage is $\Delta V_g \approx 5.7$ V. Inset: FFT spectrum of $\Delta G$ vs. $V_g$ at $B = 0.6$ V. The peak is located at period $\Delta V_g = 5.7$ V. **c** Total conductance $G$ vs. gate voltage at zero magnetic field. Inset: By comparing to another device with identical geometry (Device 2), the Dirac point is expected to be at $V_{g0} \approx -113$ V

conductance traces at $\Phi = 1.5\ \Phi_0$ and $2\ \Phi_0$ is out of phase, as expected from TI surface states (Fig. 2b)[12–14]. The periodicity estimated from the FFT (Fig. 2b, inset) is $\Delta V_g \approx 5.7$ V, which corresponds to the level spacing between neighboring transverse modes $\Delta = \frac{hv_F}{R} \cong 4.7$ meV. In Fig. 2c, the total conductance difference between $V_g = -32$ V and $-20$ V, corresponding to a chemical potential difference of $2\ \Delta$, is ~2.5 ($e^2/h$). Thus, we estimate that the chemical potential is near $\mu \approx 24\ \Delta$ when total conductance $G \approx 30$ ($e^2/h$), indicating that our experiments are executed far away from the conduction band edge (~50 $\Delta$)[9,23], and so bulk-state contributions are not considered in our analysis.

**AB oscillations in the mechanical resonant frequency**. The 1D sub-band model of a topological insulator nanowire predicts periodic oscillations of quantum capacitance and conductance with respect to both Fermi energy and Aharonov–Bohm phase. In our device structure, the strong electro-mechanical coupling converts the quantum capacitance oscillation into mechanical resonance shifts; measurements of mechanical resonance and conductance confirm these effects of the surface states in the Bi$_2$Se$_3$ nanowire. We provide a comprehensive experimental and numerical presentation of the mechanical resonant frequency shift in Fig. 3. The AB oscillations of resonant frequency at different gate voltages are shown in Fig. 3a: as the chemical potential approaches the Dirac point from $V_g = -18.8$ V to $-31.2$ V, an increase in oscillation amplitude is noticeable, as expected from Fig. 1h. The FFT (Fig. 3a, inset) clearly shows that the period of resonant frequency oscillation equals that of conductance, $\Delta B = 0.4$ T. A phase alternation of resonant frequency oscillation takes place with gate voltages as well. The mechanical resonant frequency shift vs. $V_g$ curves (Fig. 3c) for integer and half-integer flux quanta also exhibit an out-of-phase relation with each other, and their oscillation period $\Delta V_g \approx 5.7$ V (Fig. 3c, inset) agrees well with conductance oscillation data. We numerically compute the shift of resonant frequency for an extended energy window in Fig. 3b. For the modulation of mechanical resonant frequency shift, we employ a quasi-1D model of a Dirac fermion with complete eigenenergy information for a given magnetic flux from Eq. (3). We first obtain the DOS from the Green's function and then compute the shift of resonant frequency according to Eq. (5). A scattering time of $\tau = 0.7$ ps in the DOS computation is found

to explain our data most satisfactorily. We observe a qualitative change in the oscillation pattern from a checkerboard-like shape for $\mu > 25\ \Delta$ where $\Delta k_I$ in Eq. (5) is larger, to a diamond-like shape for $\mu < 25\ \Delta$ where $\Delta k_{II}$ begins to dominate. This crossover must take place at a certain energy in the measurement of 2D Dirac fermions as the chemical potential approaches the Dirac point, where the quantum capacitance holds the majority of potential difference (Eq. (2)). Not only does the overall shape of the pattern change, but the relative correlation with respect to conductance modulation reverses; for example, note the location of peaks and dips at $\Phi = 0$ along the energy. From total conductance measurements, we estimate the chemical potential to be ~24 $\Delta$, which is near the crossover energy. In the chemical potential range $\mu \cong 23\ \Delta$ to $25\ \Delta$, further comparisons between experiment and theory along magnetic flux confirms the validity of our analysis (Fig. 3e–h).

## Discussion
The characterization of various topological phases of matter is at the cutting edge of experimental condensed matter physics. We employ nanomechanical resonance measurements to characterize the topological phase of a Bi$_2$Se$_3$ nanowire through AB-type oscillation. The mechanical resonant frequency of the TI nanowire reveals its density of states via quantum capacitance effects, with a DOS-derivative over energy identified to dominate the shifts in resonant frequency. Aided by the simultaneous measurements of resonant frequency and conductance, we discover that the related spectral feature of the quasi-1D modes in the topological surface states is well reflected in the resonant frequency shift, and that a more pronounced shift occurs as the chemical potential approaches the Dirac point. Our results suggest that the nanomechanical detection scheme presented here could be generally applicable to a variety of materials with Dirac electronic structures, and also that the technique could provide a novel route to studying the physics of the mechanical motion of a nanostructure coupled to its non-trivial electronic states.

## Methods
**Device fabrication**. Single-crystalline Bi$_2$Se$_3$ nanowires are synthesized by chemical vapor deposition using Au nanoparticles as a catalyst. A single Bi$_2$Se$_3$ nanowire is transferred with a nano-manipulator to a SiO$_2$/Si substrate with a trench having a bottom-gate electrode. For trench fabrication, poly(methyl methacrylate) (PMMA)

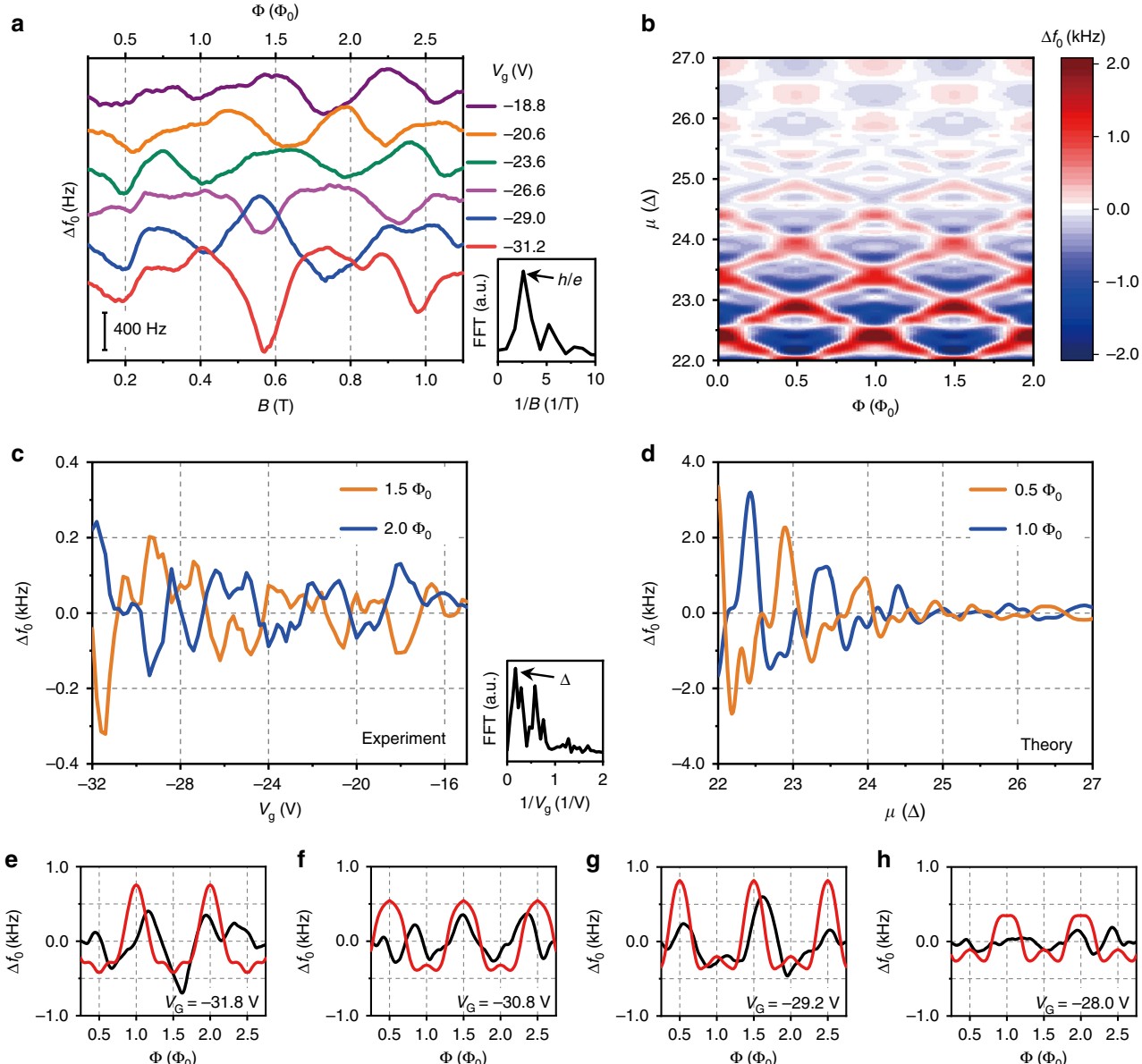

**Fig. 3** AB oscillations in the mechanical resonant frequency of a TI nanowire. **a** Measured mechanical resonance shift $\Delta f_0$ as a function of magnetic field at different gate voltages. The $\Delta f_0$ oscillates with a period of $\Delta B = 0.4$ T as identified from the first peak in the FFT spectrum of $\Delta f_0$ vs. $B$ (inset), identical to the period observed in conductance measurements. Inset: FFT spectrum of $\Delta f_0$ vs. $B$ at $V_g = -26.6$ V. Arrows indicate the peak position at period $\Delta B = 0.4$ T. **b** Model calculations of mechanical resonant frequency shift plotted as a function of chemical potential $\mu$ and magnetic flux $\Phi$ showing a change in oscillation pattern. **c, d** Measured (**c**) and calculated (**d**) resonant frequency shift as a function of gate voltages $V_g$ and chemical potential $\mu$, respectively, at half-integer (orange) and integer (blue) flux quanta. Inset in **c**: FFT spectrum of $\Delta f_0$ vs. $V_g$ at $B = 0.6$ T. The peak is located at period $\Delta V_g = 5.7$ V, identical to the period observed in conductance measurements. **e–h** Calculated (red) and measured (black) resonant frequency shifts as a function of magnetic flux at gate voltages $V_g = -31.8$ V (**e**), $-30.8$ V (**f**), $-29.2$ V (**g**), and $-28.0$ V (**h**). Corresponding chemical potentials are 23.53, 23.82, 23.91, and 24.33, respectively, in the units of $\Delta$.

resist is spin coated onto the SiO$_2$/Si substrate and the trench area is patterned using electron beam lithography followed by etching the 500 nm-thick-SiO$_2$ layer with buffered oxide etchant (BOE). The bottom-gate electrode is deposited at the bottom of the trench with electron beam evaporation (5 nm Ti, 45 nm Au). After transferring the nanowire, a PMMA resist is spin coated onto the nanowire and baked at 180 °C for 2 min. Selected areas around the nanowire are patterned using electron beam lithography and 20 nm/200 nm Ti/Au electrodes are deposited via AC-sputtering. For metallic contacts to the nanowire, PMMA residue and the native oxide layer on the nanowire surface is removed using a plasma asher and immersing in BOE for 10 s before sputtering. A gate electrode is located $d \sim 170$ nm below the suspended nanowire, and source and drain contacts are provided to compose a field-effect transistor geometry. The suspended nanowire has dimensions of width 105 nm, thickness 116 nm, and length 1.5 μm.

**Measurements**. The device is measured in a $^3$He/$^4$He dilution refrigerator, and typical measurements are performed at 50 mK. The gate electrode is biased with DC voltage to change the chemical potential of the nanowire, with a sweeping magnetic field along the nanowire direction to modulate the flux through the wire cross-section. Electrical properties of the nanowire are examined by monitoring electrical conductance as measured by comparing near-DC (~17 Hz) source-drain current and voltage across the nanowire. The suspended Bi$_2$Se$_3$ nanowire behaves as a mechanical resonator in response to external force[27]. To detect its fundamental mode, RF voltage is applied to the gate in addition to DC, and the resulting RF current due to gate-capacitance modulation is monitored (Fig. 1b)[18,20]. An example of measured amplitude near one mechanical resonance is plotted in the inset of Fig. 1c. The typical mechanical resonant frequency is about 115 MHz, and the quality factor is about $1.2 \times 10^4$.

**Quasi-1D mode DOS**. The density of states is computed by taking the imaginary part of the retarded Green's function,

$$\nu(E) = -\frac{1}{\pi} Tr \, Im\left[\frac{1}{E + i\eta - H}\right],$$

$$= \sum_{l,n=-\infty}^{\infty} -\frac{1}{\pi} Im\left[\frac{1}{E + i\eta - \varepsilon(n, k = \frac{2\pi}{L}l, \Phi)}\right],$$

where the dispersion relation $\varepsilon(n, k, \Phi)$ in Eq. (3) is used. Index $n$ indicates the transverse modes, and $l$ indicates the eigenmodes within the same transverse mode. $\eta$ is a measure of energy broadening caused by elastic scattering from impurities and inelastic scattering from phonons and electron–electron interaction. Thus, it is not straightforward to evaluate scattering time a priori, as only limited microscopic information on the TI nanowire is available. We find that an energy broadening of $\eta = 0.20 \, \Delta$, where $\Delta$ is the level spacing between neighboring transverse modes, explains our experimental data well, with a different choice of $\eta$ not affecting the qualitative behavior. Strictly speaking, the energy-broadening $\eta$ is an energy-dependent quantity as more scattering channels are available at higher chemical potential. Nevertheless, since our measurement is carried out within a small energy window ($\sim$2 $\Delta$), we ignore changes in $\eta$. Instead, for the calculation of resonant frequency shift we made an average over $\mu_0$ as it pertains to larger energy uncertainty. For illustrative purposes, we set $2\pi R/L \ll 1$ and $\eta = 0.05 \, \Delta$ in Fig. 1e–h, while we set $2\pi R/L = 1/3$ and $\eta = 0.20 \, \Delta$ in Fig. 3b and Supplementary Fig. 12c for a realistic description of the experiment.

**Lattice model conductance**. Landauer–Buttiker formalism[28] is employed to compute conductance in the quasi-1D modes on the surface of a 3D lattice model for TIs. According to Fisher and Lee[29], DC conductance $G$ of a finite system with static disorder is related to its transmission matrix $t$ by the following relation:

$$G = \frac{e^2}{h} Tr(t^\dagger t),$$

which is the sum of transmission eigenvalues. This expression is valid for any number of scattering channels. The transmission coefficient is then computed from the Green's function in real space representation by taking an element connecting the location of the left lead to that of the right lead. For a memory efficient recursive Green's function method, see the work by Conan[30].

The lattice Hamiltonian[31] employed is

$$H = -t \sum_{n,j=1,2,3} \left(\psi_{n+\hat{e}_j}^\dagger \frac{\Gamma^1 - i\Gamma^{j+1}}{2} \psi_n + \text{h.c.}\right) + \sum_n \psi_n^\dagger (m\Gamma^1)\psi_n + \sum_{n \in \text{surface}} \psi_n^\dagger (V_n \Gamma^0)\psi_n,$$

where the gamma matrices are $\Gamma^{(1,2,3,4)} = (I \otimes s_z, -\sigma_y \otimes s_x, \sigma_x \otimes s_x, -I \otimes s_u)$, where $s$ and $\sigma$ are Pauli matrices referring to orbital and spin space, respectively, $\Gamma^0$ is the identity matrix, and $t = 1$ and $m = 1.8$ are used. The first term constitutes nearest-neighbor hopping in the lattice, and the second term provides a constant mass. The third term is onsite random potential with uniform distribution, $V_n \in [-W, W]$. A disorder strength of $W = 4.5 \, \Delta$ is used for Supplementary Fig. 12d (see Supplementary Note 1 for conductance maps at other disorder strengths, $W = 1.5 \, \Delta$, 3.0 $\Delta$, and 6.0 $\Delta$). The open boundary condition is introduced in $x$-direction and $z$-direction with lattice size $N_x = N_z = 8$, and the direction of current $\hat{y}$ is chosen without the loss of generality. Impurities are introduced only on the open surface to minimize coupling to bulk modes, as our interest is the transport in quasi-1D surface modes in the presence of disorder. To match the ratio between the circumference and length of the TI nanowire in experiment, we set the length of the disordered region $N_y = 96$ to be connected to two semi-infinite leads. Magnetic flux $\Phi$ threading through the nanowire is added in the lattice model by replacement hopping along the $x$-direction as

$$\psi_{n_x+\hat{e}_x, n_y, n_z}^\dagger (H_x) \psi_{n_x, n_y, n_z} \rightarrow \psi_{n_x+\hat{e}_x, n_y, n_z}^\dagger \left(H_x e^{\frac{i2\pi n_z}{N_x N_z}\frac{\Phi}{\Phi_0}}\right) \psi_{n_x, n_y, n_z},$$

which introduces a uniform magnetic field. Because boundary modes have finite penetration depths into the bulk, AB oscillation periodicity is larger than $\Phi = \Phi_0$, while periodicity is also dependent on energy as eigenmodes near the bulk band are more delocalized than on the open surface.

## Data availability

The data that support the findings of this study are available from the corresponding author upon reasonable request.

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

## Acknowledgements

This work was supported by the Basic Science Research Program through National Research Foundation of Korea (NRF) funded by the Ministry of Science and ICT (2016R1C1B2014713, 2016R1A5A1008184), and Korea Research Institute of Standards and Science (KRISS-2018-GP2018-0017). K.W.K. acknowledges financial support from Institute for Basic Science (IBS-R024-D1).

## Author contributions

M.K., J.K., and J.S. conceived the experiments. M.K. and J.K. fabricated the devices and performed the experiments. Y.H., D.Y., Y.-J.D., and B.K. helped in sample fabrication and characterization. K.W.K. performed theoretical modeling. M.K., K.W.K., and J.S. analyzed the data and prepared the paper. J.S. supervised the project. All the authors contributed to the discussions and paper preparation.

## Competing interests

The authors declare no competing interests.
