## [Peer Review File · Nature Communications]

Reviewers' comments:

Reviewer #1 (Remarks to the Author):

This manuscript reports on both quantum electron transport and electro-mechanical measurements on a topological insulator nanowire. The authors provide convincing data that support the modulation of the density of states due to the Aharonov-Bohm effect in a topological insulator nanowire. This is a nice result, since it is not trivial to extract the density of state from conductance measurements. This work is interesting for the nano-mechanics community, since it shows how electro-mechanics can be used to reveal an exciting effect in the quantum electron transport of a topological insulator, a system that is attracting considerable attention. I recommend publication in Nature Communications. I have the following questions/comments to the authors.

1-In Fig 2a, what is the peak of conductance at zero B field? Should ΔG not be negative for the orange and blue curves?

2- The data in Fig. 4a and Fig. 4c convincingly show the AB effect. By contrast, the data in Fig. 3a,b are not convincing. Why do the authors show these data.

3-The G trace in Fig. 3b is different from that in Fig. 2b. Why?

4-The manuscript is overall written in a OK way. But there are several sentences that are vague. For instance, the sentence in the abstract (21-23) is unclear. What is the difference between Refs. 12-14 and Ref. 15

5-It is difficult to see the quadratic dependence of conductance versus gate voltage in the inset of Fig.2 c, as stated in the text (103,104).

Reviewer #2 (Remarks to the Author):

In the paper titled "Nanomechanical characterization of quantum interference in a topological insulator nanowire", Kim et al. studied AB oscillations in a topological insulator wire by exciting the wire nanomechanically. Similar experiment has been done previously as reported in Ref. 13, but the nanomechanical part is new. The sensitivity of frequency shifts versus density of state change in the nanowire looks interesting, which represents a novel coupling between nanomechanics and topological physics. While I recognize the novelty of work, which I think meets the broad expectation of Nature Communications, there are issues I want the authors to clarify before making any recommendation to publication.

1) What transport regime? The 1D subband model is an essential assumption of this work. The data shown suggest it is certainly not ballistic since conductance quantization is missing. Instead it seems highly disordered since the disorder potential goes to several Δ . Could the authors clarify in the manuscript? To my impression, the disorder can not be too high to see AB oscillations like Ref. 12 has pointed out.

2) Spin-orbit coupling. My first expectation of a nanomechanical topological wire, after seeing the title, was to see the coupling between spin sub-band transport and nanomechanical motion mediated by the strong spin-orbit coupling in the topological insulator. However I didn't see this happening in this work. There should be a sizable spin-orbit coupled term in Equation (4), which modifies the energy of a

particular spin subband in external magnetic field. Could the authors comment?

3) Quality of data. The authors did show some nice conductance oscillations when sweeping the magnetic field, which shows the characteristic h/e period. However some are not, e.g. the frequency shifts dependent of gate voltages and magnetic flux. The out-of-phase relationship does not look very convincing in Fig. 3a, Fig. 3b, 4c and 4d. Since the authors showed the simulations of 2d mapping of frequency shifts and conductance change dependent on chemical potential and flux in Fig. 3c, 3d and 4b, I strongly urge the authors to show the corresponding 2D experimental data plot. It would be much more convincing if the overall patterns match when data quality is an issue.

4) Fig. 3 and Fig. 4 seem to talk about the same content, even the titles of figures are nearly the same. They can be combined for the sake of clarity.

Overall, I think it is an interesting manuscript but some issues need to be resolved first in order to get published in Nature Communications.

We appreciate the Reviewers for their valuable comments to improve our manuscript. All changes to the manuscript and responses to the Reviewers' comments are as follows:

Reply to Reviewer 1

This manuscript reports on both quantum electron transport and electro-mechanical measurements on a topological insulator nanowire. The authors provide convincing data that support the modulation of the density of states due to the Aharonov-Bohm effect in a topological insulator nanowire. This is a nice result, since it is not trivial to extract the density of state from conductance measurements. This work is interesting for the nano-mechanics community, since it shows how electro-mechanics can be used to reveal an exciting effect in the quantum electron transport of a topological insulator, a system that is attracting considerable attention. I recommend publication in Nature Communications. I have the following questions/comments to the authors.

Question 1) In Fig 2a, what is the peak of conductance at zero B field? Should ΔG not be negative for the orange and blue curves?

Answer) The peak of conductance at zero B field is the signature of weak anti-localization, which is observed due to the strong spin-orbit coupling in topological insulator nanowire [1,2]. Although the conductance AB oscillation should show a dip at zero B field for orange ($V_g = -19.5$ V), magenta ($V_g = -25.8$ V), and blue ($V_g = -31.8$ V) curves, the weak anti-localization causes the conductance oscillation peaks.

[1] Peng, H. *et al.* Aharonov-Bohm interference in topological insulator nanoribbons. *Nat. Mater.* **9**, 225–229 (2010).

[2] Hikami, S., Larkin, A. I. & Nagaoka, Y. Spin-orbit interaction and magnetoresistance in the two dimensional random system. *Prog. Theor. Phys.* **63**, 707–710 (1980).

Revision) Following the discussion of AB conductance oscillation data, in the revised manuscript we have added: “*Note that the pronounced peak at $B = 0$ is the signature of weak anti-localization^{13,26}.*”

Question 2) The data in Fig. 4a and Fig. 4c convincingly show the AB effect. By contrast, the data in Fig. 3a, b are not convincing. Why do the authors show these data.

Answer) Besides the AB oscillation of the frequency shift over the external magnetic field, the correlation of conductance and frequency shift over energy is an independent set of information that characterizes our device.

In Figure 1e and 1f, we show the overall behavior of conductance, the electronic DOS, and the frequency shift Δf_I and Δf_{II} . Because two contributions with different functional forms are expected in the mechanical resonant frequency shift (Δf_I and Δf_{II}), we previously showed the correlation between conductance and frequency shift over the energy in Figure 3 in order

to identify the device working regime, and finally Figure 4 showed more comprehensive information of the frequency shift. Nevertheless, we agree that Figure 4 shows convincing AB oscillations and Figure 3 is not necessary for the conclusion of paper. We have made appropriate changes in the manuscript.

Revision) For better presentation of our results, Figure 3 has been moved to Supplementary Note 7, and Figure 4 is now Figure 3.

Question 3) The G trace in Fig. 3b is different from that in Fig. 2b. Why?

Answer) We performed the G versus V_g measurement twice. One measurement was executed with conductance measurement alone and the other with mechanical resonance measurement simultaneously. The two measurement results are shown in Fig. 2b and Fig. 3a–b respectively. The overall conductance traces in Fig. 2b and Fig. 3a–b are consistent with each other, as shown in Fig. R1 below, verifying the data reliability.

Figure R1. (a) Conductance plot of Fig. 2b and Fig. 3a at half-integer flux quanta. (b) Conductance plot of Fig. 2b and Fig. 3b at integer flux quanta.

Revision) As stated in the response to Question 2, Figure 3 has been moved Supplementary Note 7 and Figure 4 is now Figure 3.

Question 4) The manuscript is overall written in a OK way. But there are several sentences that are vague. For instance, the sentence in the abstract (21-23) is unclear. What is the difference between Refs. 12-14 and Ref. 15.

Answer) We have re-phrased the sentence (21-23) in the introduction and combined Refs. 12-14 and Ref. 15.

Revision) In the introduction, we now write “*In TI nanowires, the gapless surface states exhibit Aharonov–Bohm (AB) oscillations in conductance due to a change in the number of transverse one-dimensional (1D) modes in transport*¹²⁻¹⁵.”

Question 5) It is difficult to see the quadratic dependence of conductance versus gate voltage

in the inset of Fig.2 c, as stated in the text (103,104).

Answer) The relation of chemical potential and gate voltage $(\mu - \mu_0) \sim V_g^2$ should be corrected to $(\mu - \mu_0) \sim V_g^{1/2}$, and the $(\mu - \mu_0) \sim V_g^{1/2}$ relation is shown in Figure R2 (Supplementary Figure 5 in Supplementary Information). Thus, the total conductance has a relation of $G \sim V_g^{1/2}$ and the fitting results are shown in Figure R3.

Figure R2. Calculated relation of gate voltage V_g and chemical potential μ (Supplementary Figure 5).

Figure R3. Total conductance G vs. gate voltage with $G \sim \sqrt{V_g}$ fit result.

Revision) We decided that the explanation about conductance versus gate voltage is not necessary for the main point of our manuscript. Thus, we have removed the following sentences: “Since the DOS changes over energy, gate voltage is not linearly proportional to chemical potential change, with approximately $\mu - \mu_0 \sim V_g^2$ for 2D Dirac dispersion. Thus, change in total conductance is also a quadratic function of gate voltage, as the chemical potential of the TI nanowire approaches the Dirac point (Fig. 2c, inset).”

Reply to Reviewer 2

In the paper titled "Nanomechanical characterization of quantum interference in a topological insulator nanowire", Kim et al. studied AB oscillations in a topological insulator wire by exciting the wire nanomechanically. Similar experiment has been done previously as reported in Ref. 13, but the nanomechanical part is new. The sensitivity of frequency shifts versus density of state change in the nanowire looks interesting, which represents a novel coupling between nanomechanics and topological physics. While I recognize the novelty of work, which I think meets the broad expectation of Nature Communications, there are issues I want the authors to clarify before making any recommendation to publication.

Question 1) What transport regime? The 1D subband model is an essential assumption of this work. The data shown suggest it is certainly not ballistic since conductance quantization is missing. Instead it seems highly disordered since the disorder potential goes to several Δ . Could the authors clarify in the manuscript? To my impression, the disorder can not be too high to see AB oscillations like Ref. 12 has pointed out.

Answer) When the linewidth broadening by elastic scattering source (static impurities) is larger than the level spacing between neighboring 1D sub-bands, the magneto-conductance shows AAS oscillation instead of AB oscillation. In experiment, because the periodicity that we observed is h/e , we believe our device is closer to ballistic rather than diffusive in transport. Nevertheless, the magnitude of conductance oscillation is smaller than the conductance unit, which is probably due to incoherent phonon scatterings. In fact, other previous experiments also reported the magnitude of AB oscillation to be smaller than the conductance unit [1,2].

Regarding the conductance calculation, the linewidth broadening involves the electronic DOS as well as the disorder strength. For disorder strength $W = 4.5\Delta$ used in the transport calculation, the linewidth broadening is approximately $\sim 0.5\Delta$, which puts the system in the ballistic regime (please see Supplementary Note 6 for details).

[1] Jauregui, L. A., Pettes, M. T., Rokhinson, L. P., Shi, L. & Chen, Y. P. Magnetic field-induced helical mode and topological transitions in a topological insulator nanoribbon. *Nat. Nanotechnol.* **11**, 345–351 (2016).

[2] Hong, S. S., Zhang, Y., Cha, J. J., Qi, X. L. & Cui, Y. One-dimensional helical transport in topological insulator nanowire interferometers. *Nano Lett.* **14**, 2815–2821 (2014).

Revision) For reader clarity, we have rewritten and added to the following discussion about the nature of transport in the revised manuscript. “As indicated in the fast Fourier transformation (FFT, Fig. 2a, inset), the magneto-conductance ΔG oscillates with a period of $\Delta B = 0.4 T$, which is consistent with the expected period, $\Phi_0/S = 0.37 T$ with nanowire cross-section area S (Supplementary Note 1). There is only a small fraction of Altshuler–Aronov–Spivak oscillation with $\Phi_0/2$ period, indicating that the nature of transport is close to ballistic, rather than diffusive; the fact that the AB oscillation amplitude is smaller than the

conductance unit could originate from the presence of incoherent scatterings^{15,25}.”

Question 2) Spin-orbit coupling. My first expectation of a nanomechanical topological wire, after seeing the title, was to see the coupling between spin sub-band transport and nanomechanical motion mediated by the strong spin-orbit coupling in the topological insulator. However I didn't see this happening in this work. There should be a sizable spin-orbit coupled term in Equation (4), which modifies the energy of a particular spin subband in external magnetic field. Could the authors comment?

Answer) The magnetic field is applied in-plane to the Dirac electrons. And, its incorporation is precisely done in the continuum limit in Equation (3). Thus, spectral modification in our spin-orbit coupled Dirac fermion is well considered. Then, when it comes to the energy relation of the electric-mechanical circuit in Equation (4), it is only important to consider the DOS regardless of spin types.

It is certainly a very interesting idea to have spin-dependent coupling between mechanical oscillation and Dirac surface states. If the coupling is strong enough, the nanomechanical device would be able to probe not only electronic DOS but spin-dependent information as well. Unfortunately, in the current setup of our device, the spin-dependent energy splitting via Rashba spin-orbit coupling is expected to be vanishingly small by the following estimate.

The external electric field difference caused by the mechanical motion is

$$\Delta F_{ext} \approx \frac{Vx}{d} = \frac{(30V)(265fm)}{(100nm)^2} = 795 (V/m).$$

The Rashba coupling in Bi₂Se₃ comes from the electric field induced by the band bending potential near the open surface, and it can be estimated as (Ref: PRL **107**, 096802),

$$\Delta F_{int} \approx \frac{V}{d} = \frac{(0.3V)}{(100\text{\AA})} = 3 \times 10^7 (V/m).$$

Therefore, the fraction of electric field change experienced by the TI nanowire during the mechanical oscillation is

$$\frac{\Delta F_{ext}}{\Delta F_{int}} \approx 2.65 \times 10^{-5}.$$

The experimentally known Rashba spin-orbit coupling strength on the open surface is $\alpha_{int} \approx 0.1 (eV\text{\AA})$. The estimate above yields the Rashba coupling induced by the mechanical motion as $\alpha_{ext} \approx 2.65 \times 10^{-6} (eV\text{\AA})$, which is substantially smaller than the intrinsic value. Then, the modification on the surface Dirac Hamiltonian is written as,

$$H = v_k(k_x\sigma_y - k_y\sigma_x) + \alpha_{ext}(\vec{k} \times \vec{z}) \cdot \vec{\sigma}.$$

The Fermi velocity $v_k = 3.55 (eV\text{\AA})$ (Ref: PRL 105, 076802) is significantly larger than the Rashba coupling induced by the mechanical motion. In this regard, in the electro-mechanical coupling Equation (4), we can safely ignore the extrinsic Rashba effect.

Question 3) Quality of data. The authors did show some nice conductance oscillations when sweeping the magnetic field, which shows the characteristic h/e period. However some are not, e.g. the frequency shifts dependent of gate voltages and magnetic flux. The out-of-phase relationship does not look very convincing in Fig. 3a, Fig. 3b, 4c and 4d. Since the authors showed the simulations of 2d mapping of frequency shifts and conductance change dependent on chemical potential and flux in Fig. 3c, 3d and 4b, I strongly urge the authors to show the corresponding 2D experimental data plot. It would be much more convincing if the overall patterns match when data quality is an issue.

Answer) 1. For the mechanical resonant frequency shift, the AB oscillation is expected to be diminished in the transition region, where Δf_I and Δf_{II} change their relative contributions in the total frequency shift, and our experiments are performed across this region (Fig. 4b; Fig.3b in the revised manuscript, also refer to Supplementary Note 5). Thus, the conductance and frequency shift show an out-of-phase relation in the range of $V_g = -32$ V to -26.6 V but their relative relation becomes unclear in the range of $V_g = -26.6$ V to -15 V near the transition region. Nevertheless, the existence of AB oscillation in the resonant frequency is supported clearly by the h/e peaks in the FFT analysis of the data shown at the sub-band gap Δ (Fig. 4a; Fig. 3a in the revised manuscript).

2. We compare the 2D experimental data and simulation maps of mechanical resonant frequency shifts in Figure R4. The repeating diamond pattern is seen in both maps (Figure R4a and b), with a good match between them as shown in Figure R4c. To compare the pattern, we extract the selected Fourier component (Figure R4e and f) in the experimental data and normalize the amplitude in both experimental data and simulation maps. The mechanical resonant frequency shift versus V_g curves (Figure R4d) for integer and half-integer flux quanta also show a clear out-of-phase relation with each other after extracting the selected Fourier component (Figure R4e and f).

Figure R4. (a, b) Normalized 2D experimental data (a) and simulation color map (b) of mechanical resonant frequency shifts. (c) Overlaid plot of experimental data in (a) with the simulation color map in (b). (d) Mechanical resonant frequency shift as a function of gate voltages at half-integer and integer flux quanta showing out-of-phase relation. (e, f) The Fourier component slower than $1.5 h/e$ period in flux (e) and the Fourier components within a 20 % window for the major oscillation Δ period in energy (f) are selected to plot the experimental data in (a) and (d).

Revision) For better presentation of our results, Figure 3 has been moved to Supplementary Note 7, and Figure 4 is now Figure 3 in the revised manuscript. We have also added the above Figure R4 as Supplementary Figure 13 in Supplementary Note 8.

Question 4) Fig. 3 and Fig. 4 seem to talk about the same content, even the titles of figures are nearly the same. They can be combined for the sake of clarity.

Answer) Please see our response to Question 2 from Reviewer 1, as follows. Besides the AB oscillation of the frequency shift over the external magnetic field, the correlation of conductance and frequency shift over energy is an independent set of information that characterizes our device.

In Figure 1e and 1f, we show the overall behavior of conductance, the electronic DOS, and the frequency shift Δf_I and Δf_{II} . Because two contributions with different functional forms are expected in the mechanical resonant frequency shift (Δf_I and Δf_{II}), we previously showed the correlation between conductance and frequency shift over the energy in Figure 3 in order to identify the device working regime, and finally Figure 4 showed more comprehensive information of the frequency shift. Nevertheless, we agree that Figure 3 is not necessary for the conclusion of paper. We have made appropriate changes in the manuscript.

Revision) For better presentation of our results, Figure 3 has been moved to Supplementary Note 7, and Figure 4 is now Figure 3 in the revised manuscript.

Reviewers' comments:

Reviewer #1 (Remarks to the Author):

The authors answered my comments in a convincing way. This a very important work that shows for the first time how nanomechanics can be used to reveal the electron transport properties of a topological insulator. This manuscript is interesting for both the nanomechanics community and the topological insulator community. I recommend publication without hesitation.

Reviewer #2 (Remarks to the Author):

In my first round review I raised a few questions in the manuscript regarding to the transport regimes, the role of spin orbit coupling and the data quality. I can see the authors have put significant efforts in addressing my points and I am partially satisfied by their response. For example, both reviewers seem to have issues with the data quality. I asked the authors to plot 2D data with side by side comparison of their simulations. They did include this in Fig. R4. I can not say it is a good match between experiment and simulation, but I guess I can let it go as some features do follow each other.

I am not satisfied with the explanation of spin-orbit coupling term in the Hamiltonian. The authors estimated the external electric field difference caused by mechanical motion in ONE DIRECTION. I disagree since they shouldn't use the static V_g to estimate the field difference, rather they should use the RF field. And the max field difference happens when the nanomechanical resonator reverses its direction. One can naively think the spin and momentum lock each other in the resonator (since it is a topological insulator), once the resonator reverses direction, the spin should flip and shift the energy caused by the Zeeman effect in a magnetic field.

Anyway, I like this paper in some degree but I am afraid I have to request further comments from the authors.

We appreciate the Reviewers for their valuable comments to improve our manuscript. All changes to the manuscript and responses to the Reviewers' comments are as follows:

Reviewer #1 (Remarks to the Author):

The authors answered my comments in a convincing way. This a very important work that shows for the first time how nanomechanics can be used to reveal the electron transport properties of a topological insulator. This manuscript is interesting for both the nanomechanics community and the topological insulator community. I recommend publication without hesitation.

Reviewer #2 (Remarks to the Author):

In my first round review I raised a few questions in the manuscript regarding to the transport regimes, the role of spin orbit coupling and the data quality. I can see the authors have put significant efforts in addressing my points and I am partially satisfied by their response. For example, both reviewers seem to have issues with the data quality. I asked the authors to plot 2D data with side by side comparison of their simulations. They did include this in Fig. R4. I can not say it is a good match between experiment and simulation, but I guess I can let it go as some features do follow each other.

I am not satisfied with the explanation of spin-orbit coupling term in the Hamiltonian. The authors estimated the external electric field difference caused by mechanical motion in ONE DIRECTION. I disagree since they shouldn't use the static V_g to estimate the field difference, rather they should use the RF field. And the max field difference happens when the nanomechanical resonator reverses its direction. One can naively think the spin and momentum lock each other in the resonator (since it is a topological insulator), once the resonator reverses direction, the spin should flip and shift the energy caused by the Zeeman effect in a magnetic field.

Anyway, I like this paper in some degree but I am afraid I have to request further comments from the authors.

Answer) We appreciate the Reviewer's valuable comments that have improved the manuscript and also our understanding. We examine here the possibility of having a Zeeman-type spin splitting by the threaded magnetic field B (~ 1 T) into the nanowire. We find that, associated with the motion of the resonator, which is driven by the radio-frequency AC voltage source, spin polarization arises on the sides of the nanowire, and that the polarization can couple to the external magnetic field. However, the effect turns out small enough to be neglected. In short, it originates from the interplay between the external AC electric field and the relaxation of helical electronic states. The detailed calculation is as follows.

Figure R1. Schematic representation of the topological insulator nanowire device.

In the figure above, the direction of the external magnetic field is out of plane, $\vec{B} = B\hat{x}$. The effective Hamiltonian of Dirac surface states is specified assuming that the penetration depth into the bulk is smaller than the size of nanowire. A time-dependent electric field is applied from the gate to the nanowire direction, $E_z(t) = V_{\text{RF}}\sin(2\pi f_{\text{RF}}t)/d$ with frequency $f_{\text{RF}} = 115$ MHz and $V_{\text{RF}} = 0.4$ mV. On the other hand, the relaxation time of electrons is $\tau \sim 1$ ps. This implies that the effective external electric field is reduced as

$$E_{\text{eff}}(t) \approx \tau \frac{dE_z(t)}{dt} = 2\pi f_{\text{RF}}\tau \frac{V_{\text{RF}}\cos(2\pi f_{\text{RF}}t)}{d},$$

due to screening by electrons. Note that the vibration amplitude of the nanowire (~ 265 pm) is negligible compared to d .

The effective electric field drives the electron population out of equilibrium. According to the Boltzmann equation:

$$\begin{aligned} \delta f_{k_z} &= f_{k_z} - f_0 \\ &= e\tau E_{\text{eff}} \frac{df_{k_z}}{dp_z} \approx e\tau E_{\text{eff}} v_F \frac{df_{k_z}}{dE} \approx e\tau \left(2\pi f_{\text{RF}}\tau \frac{V_{\text{RF}}}{d} \right) v_F \frac{1}{kT} \\ &= 0.329 \end{aligned}$$

where $T = 50$ mK and $v_F = 5 \times 10^5$ m/s. In other words, on the right surface of the nanowire (see figure), there are a greater number of electrons going upward (\hat{z}) than going downward ($-\hat{z}$). Because Dirac surface states are helical, this naturally makes net spin polarization S_x , which in turn causes a spin-dependent energy shifting by the external magnetic field. Note that on the left surface, the effective surface Hamiltonian is $H_{\text{left}} = -H_{\text{right}}$, and thus the

opposite spin polarization and the total energy do not change by the Zeeman splitting.

The energy scale of the Zeeman effect under $B = 1$ T,

$$\Delta H_{\text{Zeeman}} = \frac{\mu_B g_S}{\hbar} \vec{S} \cdot \vec{B} \approx 0.058 \text{ meV}.$$

Therefore, the energy shift due to the time-dependent electric field is $\Delta E_{\text{Zeeman}} \approx 0.329 \times 0.058 \text{ meV} = 0.019 \text{ meV}$, which is more than two orders of magnitude smaller than the energy level spacing between neighboring transverse modes, $\Delta = 4.7 \text{ meV}$. As a result, we conclude that spin polarization along the magnetic field direction (\hat{x}) would appear during the relaxation of electrons in the presence of an AC external electric field, but its influence on our TI nanowire is negligibly small.

Revision) We have added the above Figure R1 as Supplementary Figure 14 in Supplementary Note 9.

REVIEWERS' COMMENTS:

Reviewer #2 (Remarks to the Author):

I am satisfied with the most recent response of the authors. The spin orbit coupling has been instrumental in SO qubit [Nature 468, 1084 (2010)] and SO coupled carbon nanotube nanomechanical oscillator [PRL, 108, 206811(2012)]. But here, due to the helical nature of electron motion and large subband spacing, the SO related energy might be truly small as the authors have pointed out. I have no further questions and recommend publication in the current form of this manuscript.

Reviewer #2

I am satisfied with the most recent response of the authors. The spin orbit coupling has been instrumental in SO qubit [Nature 468, 1084 (2010)] and SO coupled carbon nanotube nanomechanical oscillator [PRL, 108, 206811(2012)]. But here, due to the helical nature of electron motion and large subband spacing, the SO related energy might be truly small as the authors have pointed out. I have no further questions and recommend publication in the current form of this manuscript.

Answer) We are glad that our responses answered referee's questions satisfactorily. We appreciate referee's valuable comments to deepen our understanding on spin-orbit couplings in the system.